# Criteria for Diagnosis of Polycystic Ovary Syndrome during Adolescence: Literature Review

**DOI:** 10.3390/diagnostics12081931

**Published:** 2022-08-10

**Authors:** Alexia S. Peña, Ethel Codner, Selma Witchel

**Affiliations:** 1Discipline of Paediatrics, The University of Adelaide Robinson Research Institute, 72 King William Road, Adelaide, SA 5006, Australia; 2Endocrinology and Diabetes Department, Women’s and Children’s Hospital, 72 King William Road, Adelaide, SA 5006, Australia; 3Institute of Child and Maternal Research, School of Medicine, University of Chile, Santiago 836-0160, Chile; 4UPMC Children’s Hospital of Pittsburgh, University of Pittsburgh, Pittsburgh, PA 15224, USA

**Keywords:** adolescents, girls, polycystic ovary syndrome, diagnosis

## Abstract

Polycystic ovary syndrome (PCOS) is one of the most common endocrine conditions in women. PCOS may be more challenging to diagnose during adolescence due to an overlap with the physiological events of puberty, which are part of the diagnostic criteria in adult women. This review focuses on the evidence available in relation to PCOS diagnostic criteria for adolescents. Adolescent PCOS should be diagnosed using two main criteria irregular -menstrual cycles (relative to number of years post-menarche) and hyperandrogenism (clinical and/or biochemical); after excluding other conditions that mimic PCOS. Accurate definitions of the two main criteria will decrease challenges/controversies with the diagnosis and provide timely diagnosis during adolescence to establish early management. Despite the attempts to create accurate diagnostic criteria and definitions, this review highlights the limited research in this area, especially in the follow up of adolescents presenting with one diagnostic feature that are called “at risk of PCOS”. Studies in adolescents continue to use the Rotterdam diagnostic criteria that uses pelvic ultrasound. This is inappropriate, because previous and emerging data that show many healthy adolescents have polycystic ovarian morphology in the early years post-menarche. In the future, anti-Müllerian hormone levels might help support PCOS diagnosis if adolescents meet two main criteria.

## 1. Introduction

Polycystic ovary syndrome (PCOS) is one of the most common endocrine conditions, affecting 8–13% [1] of women and 3.4–19.6% of adolescent girls, depending on the diagnostic criteria used and population studied [2,3,4,5,6]. The highest prevalence (19.6%) was reported in adolescents with Type 2 diabetes [6]. PCOS is also a familial condition with twin studies indicating that hereditability is approximately 70% [7]. Positive family history of PCOS in first degree relative has been reported in 24% of adolescents with PCOS and is higher in adolescents with PCOS compared to healthy adolescents [8,9]. Over 20 genetic loci associated with PCOS were identified according to genome-wide association studies among different ethnic populations of women [10,11,12,13]. Daughters of women with PCOS have been reported to have a five-fold increased risk of developing PCOS [14].

The World Health Organization defines adolescence as the period between 10 and 19 years of age, which includes critical changes in growth, puberty, and development. These physiological changes, including menstrual irregularities, hyperandrogenism, and polycystic ovarian morphology (PCOM) overlap with adult diagnostic criteria of PCOS, making diagnosis during adolescence challenging and controversial [15,16,17,18]. The first diagnostic criteria for PCOS in adult women were established by a consensus meeting at the National Institutes of Health (NIH) in 1990 [19] and was followed by multiple consensuses, statements and/or guidelines for adult women with limited acknowledgment of the difficulties for diagnosing PCOS in adolescents [20,21,22,23]. A recent systematic review identified 13 clinical practice guidelines for diagnosis and management of PCOS with seven of those covering adolescent PCOS and highlighting the variability in the scope of the guidelines and methodologies used which may influence translation to clinical practice [24]. Over the last decade, there have been three international adolescent consensuses/guidelines supporting the use of NIH PCOS diagnostic criteria. These documents include two main diagnostic criteria: menstrual irregularities/ovulatory dysfunction and hyperandrogenism once other conditions that mimic PCOS have been excluded (Table 1) [17,19,25,26]. The 2003 Rotterdam criteria for PCOS diagnosis was not recommended in the adolescent PCOS guidelines as it is based on the presence of two of three features: menstrual irregularities, clinical or biochemical hyperandrogenism, and PCOM on ultrasound; and PCOM should not be used in adolescents [20]. The adolescent consensuses/guidelines aimed to decreased the variability on diagnosis criteria used [3,27] and highlighted the lack of longitudinal data on natural history of PCOS during adolescence [17,25,26].

There is a need for a careful approach and diagnostic criteria to provide timely diagnosis during adolescence [17,28]. Appropriate early diagnosis will enable timely management of lifelong health comorbidities associated with PCOS such as type 2 diabetes, cardio-metabolic abnormalities, non-alcoholic fatty liver disease, and psychological comorbidities; and ensure that adolescents are suitably transitioned to adult care [29,30,31]. The diagnostic criteria should avoid “over diagnosis” that causes unnecessary concerns about future fertility or other complications; and at the same time highlights the need for follow up of adolescents “at risk” of PCOS who do not fulfill the diagnostic criteria [17,25,32]. Recent quality or care improvement studies have highlighted the importance of education on adolescent PCOS diagnostic criteria to improve care [33,34]. The aim of this manuscript was to review the evidence on diagnostic criteria available for adolescents with PCOS to guide timely and appropriate diagnosis of these adolescents. This review does not include the diagnosis and management of comorbidities associated with PCOS.

## 2. Search Strategy

The following databases were searched: Ovid MEDLINE, Embase, EBM Reviews, Cochrane Central Register of Controlled Trials, EBM Reviews-Cochrane Database of Systematic Reviews, and Cumulative Index to Nursing and Allied Health Literature (CINAHL), up to March 2022. The search terms for the literature search are included in Appendix A. The search strategy followed the PRISMA model, which is shown in Figure 1. The searches performed highlighted a large number of studies in adult women, studies not relevant to PCOS diagnosis and reviews/case reports which were excluded unless they were international consensuses or evidence-based. This review included original studies in adolescents, systematic reviews and meta-analysis, population-based studies (both in selected and unselected populations), consensus papers, and international guidelines.

## 3. Main Criteria to Diagnose PCOS during Adolescence

### 3.1. Menstrual Cycle Irregularity and Ovulatory Dysfunction

Oligomenorrhea and anovulation are a cornerstone element of the diagnosis of PCOS in adult women. The first diagnostic criteria of PCOS, the NIH criteria, used oligomenorrhea/amenorrhea as a required element to diagnose PCOS in adult women [19]. A systematic review evaluating diagnostic criteria for diagnosis of PCOS during adolescence, demonstrated that almost all the studies require menstrual irregularities to be present for the diagnosis of PCOS in adolescents [16]. However, special criteria should be used to define menstrual irregularity in adolescents (Table 1) [17]. Additionally, primary amenorrhea or the lack of menstruation within three years of thelarche is a feature of adolescent PCOS within the criterion of menstrual cycle irregularities. Several studies in adult women have used the presence of anovulation as a criterion for the diagnosis of PCOS, which may be a physiologic event occurring in some menstrual cycles in the early post-menarcheal years.

In the years that follow menarche, regular menstrual cyclicity may take some time to be attained. During puberty, the gonadal axis is activated in a progressive way and the achievement of menarche does not signal a full maturation of the hormonal feedback on the hypothalamic-pituitary-ovary axis [35,36]. The frequent presence of anovulatory cycles and menstrual irregularities observed in early adolescence has been explained by the absence of the physiologic positive estrogen feedback stimulating the mid cycle luteinizing hormone (LH) surge which is required for ovulation [37]. However, immaturity in the follicle stimulating hormone (FSH) and ovarian responses have also been shown to have a role [35,36].

The American Academy of Pediatrics and American College of Obstetrics and Gynecology published criteria to define menstrual abnormalities for adolescents [38,39]. and suggested that the presence of persistent menstrual cycles longer than 45 days during the six years following menarche should be considered to be oligomenorrhea. These data were based on the fact that ninety percent of cycles are within the range of 21–45 days, and cycles longer than 90 days represent the 95th percentile for length [40,41,42]. Another element to be considered for evaluation of menstrual cyclicity in adolescents is that a higher variability in the duration of the menstrual cycles is observed in young compared to adult women [43]. However, recent studies showed that most adolescents attain regular menstrual cycles with a similar duration of adult women after two to three years post-menarche. An Italian study evaluated menstrual cycles in 3783 adolescents attending schools and showed that after 3–4 years post-menarche less than 10% of the adolescents present cycles longer than 35 days and shorter than 21 days (polymenorrhagia) [44]. Similarly, only 6% of adolescents aged 16 years showed persistent menstrual cycles longer than 35 days in a Danish cohort [45,46]. Moreover, adolescents with oligomenorrhea at the age of 15 years show a tendency to persist at the age of 18 years [47] and at the age of 26 years [48]. Based on these data, there is consensus data that there are difficulties diagnosing PCOS the years following menarche. Two international studies that reported on the diagnosis of PCOS during adolescence recommended waited two years after menarche to diagnose oligomenorrhea if persistent cycles longer than 45 days were present [25,26]. However, an international evidence-based study suggested that adolescent menstrual irregularities may be diagnosed when persistent menstrual cycles longer than 45 days, present in the 1–3 years post-menarche, and after this period the <21 days and >35 days should be used (Table 1) [17], which is similar to the criteria used in adult women [49]. In addition, if a menstrual cycle is longer than 90 days one year post-menarche, it is also a sign of menstrual irregularity (Table 1) [17].

Anovulation is another aspect that differs in adolescent girls compared to adult women. In healthy young women, only 10% of the cycles are anovulatory [50,51]. However, studies evaluating ovulation in the years following puberty have shown ovulation in only 20% of the menstrual cycles during the first year post-menarche [52], 25–35% in the second year [52,53], 45% in the fourth year [52], and reaching around 70% of the cycles between 5–9 years post-menarche [53,54]. Nevertheless, another study that evaluated ovulation in young healthy women recruited in colleges aged 16–24 years showed that one third of the cycles may be anovulatory [55]. Therefore, the determination of ovulation by serum progesterone levels in a single menstrual cycle, a method that has been used for the diagnosis of PCOS in adult women [20], is not recommended in adolescents.

Another noteworthy difference between adolescent girls and adult women is that oligomenorrhea has been used as an index of the presence of anovulation in the latter group. In adolescents, menstrual cycle irregularities do not necessarily indicate the presence of anovulation [55] and a large proportion of healthy adolescents with irregular menstrual cycles are still ovulating despite irregular and infrequent menses [56]. A similar lack of correlation between menstrual cycle duration and ovulation has been reported in adolescents with type 1 diabetes [57].

Several studies showed that the presence of oligomenorrhea in adolescents is associated with hyperandrogenism. Adolescents with oligomenorrhea (>42 days) at the age of 14 years had higher free testosterone and dehydroepiandrosterone sulfate (DHEAS) [58]. Similarly, when using the 35 days criteria for diagnosing menstrual irregularity at the age of 16 years, an evaluation of 317 Danish adolescents showed that they had higher androgen levels [45,46]. Moreover, a Finnish study that evaluated 2448 adolescents (age 16 years) showed that adolescents with oligomenorrhea had higher testosterone and free testosterone levels compared to regularly menstruating adolescents [59]. Similar data were reported in a large Dutch study that evaluated 14–16 year old adolescents [60]. Recently, it was reported that the risk of having elevated androgen levels in oligomenorrheic girls is increased in obese adolescents [61].

The presence of menstrual irregularities are associated with higher body mass index (BMI), higher blood pressure, and lower insulin sensitivity [58,62]. A prospective study showed that adolescents who have three or more menstrual cycles longer than 42 days at the age of 14 years had higher BMI, insulin, glucose levels, and insulin resistance at the age of 25 years [58], suggesting that even menstrual irregularities at a young age suggest a higher metabolic risk later as a young adult.

The importance of menstrual irregularities during adolescence as a marker of future risk of PCOS was recently reported in a long follow up study of adolescents in a Dutch cohort. Caanen et al. followed a group of 271 adolescents from the age of 15 years of age of whom 30% had oligoamenorrhea and found that the risk of developing PCOS was 22.5% in the group with oligoamenorrhea compared with 5% in the group that had regular menstrual cycles [63]. Similar data were reported in studies published in an epidemiologic Finnish study [48]. Therefore, adolescents that present with isolated irregular menstrual cycles or menstrual cycles that are not considered irregular according to time postmenarche can be defined as “at risk of PCOS” and require follow up (Figure 2).

### 3.2. Hyperandrogenism

Hyperandrogenism is typically categorized as clinical or biochemical. Hirsutism and acne are considered to be manifestations of clinical hyperandrogenism that require a comprehensive physical examination. The fact that many adolescents develop mild features of clinical hyperandrogenism during puberty confounds the diagnosis of PCOS.

The semi-objective scoring system, the Ferriman Gallwey score may be used to characterize the extent of the hirsutism but should take into account if hair removal methods have been used. The score will be affected if laser/electrolysis, waxing methods, or shaving has been used in the previous 3 months, 4 weeks, and 5 days, respectively [64]. The modified Ferriman Gallwey (mFG) score involves the assessment of nine body areas (upper lip, chin, neck, chest, upper and lower abdomen, thighs, upper and lower back) with scoring between 0–4 depending on the extent of terminal hair growth (rigid hair more than 5 mm in length). However, the optimal cut point to define hirsutism likely depends on ethnic background with higher cut offs described in Mongoloid Asian compared to White and Black women [65,66]. Of note, there are no studies defining the optimal cut off for adolescents of different ethnicities and mild hirsutism may reflect ethnic variation or normal pubertal progression rather than indicating hyperandrogenism during adolescence. Nevertheless, the cutaneous findings need to be interpreted within the clinical context of a specific patient. Based on the international evidence-based guidelines, a mFG greater than 4–6 may be consistent with hirsutism [30,67]. A cross-sectional study of 154 adolescents two years post-menarche in Canada including 60 with PCOS according to Rotterdam criteria, 48 who were classified as at risk of PCOS by authors but fulfill NIH PCOS criteria, and 46 healthy controls showed mean mFG of 17.1, 15.9, and 5.7, respectively. The presence of hirsutism and acne was similar among the adolescents with PCOS diagnosed using Rotterdam or NIH criteria [68]. Lower mean mFG scores (6–8.5) were reported in cohorts of adolescents with PCOS in the USA including Hispanics and black adolescents [69,70]. Hirsutism defined as mFG score higher than 6 and higher than 8 was reported in 60–70% and 50% of adolescents with PCOS respectively [8,68,70]. Higher hirsutism scores are related to higher testosterone levels according to population and cross-sectional studies of adolescents [59,60,69,71,72]. Hirsutism must also be distinguished from hypertrichosis, which is defined as excessive vellus hair distributed in a non-sexual pattern.

Mild to moderate acne is common among adolescents and among adolescents with PCOS [73,74]. However, when acne is more severe, PCOS should be considered to be a diagnosis [17,25,26]. There is no consensus on a single score for evaluation of the severity of acne, but in general adolescents that have larger number of comedonal lesions that are resistant to topical medications and cause scaring have severe acne. A recent systematic review and meta-analysis showed that the prevalence of acne in women with PCOS is higher compared to women without PCOS (43 vs. 21%) highlighting higher prevalence in East Asia. Additionally, this study showed that the estimated prevalence of acne was higher in adolescents with PCOS compared to women with PCOS (59 vs. 42%) [74]. Both acne and hirsutism are the most common skin manifestations of adolescents with PCOS [68,70].

Another feature of clinical hyperandrogenism is the female pattern hair loss or previously called alopecia. This is a diffuse thinning of scalp hair around the crown area that can be present in 28% of women with PCOS [75]. There are no studies specifically evaluating female pattern hair loss in adolescents with PCOS. Two studies from the Middle East including 53 and 55 adolescents with PCOS, respectively, reported only one adolescent with PCOS and female pattern hair loss (1.8%) [8,9].

In relation to biochemical hyperandrogenism, all reports stress the importance of sensitive and consistent testosterone assays. Radioimmunoassays were used to measure total testosterone but more recently, liquid chromatography-mass spectroscopy methods were developed [76]. However, reference intervals defining normative data in adolescent girls are lacking and hormone concentrations vary during the peripubertal years. An additional consideration when measuring androgen levels in any woman is that should be in the absence of hormonal contraception for at least three months to avoid interference with results.

Nicolaides and colleagues recommended use of free testosterone using a reliable assay, free androgen index, and bioavailable testosterone as measure of biochemical hyperandrogenism [77]. Specific recommendations for total testosterone concentration range from 55 ng/dL (1.9 nmol/L) [25]. Khashchenko et al. reported androgen concentrations in 130 adolescents aged 15–17 years and two years post-menarche diagnosed with PCOS by Rotterdam criteria. Median hormone concentrations were testosterone 55 ng/dL (1.9 nmol/L) (range 35–72 ng/dL (1.2–2.5 nmol/L)) and androstenedione concentrations 15.8 ng/mL (55.2 nmol/L) (range 11.6–23.3 ng/mL (40.5–81.3 nmol/L)). For DHEAS, the mean± standard deviation was 6.8 ± 3.2 µmol/L [78]. These investigators determined that using cut-points for testosterone > 33 ng/dL (1.15 nmol/L), androstenedione > 11.45 ng/mL (40 nmol/L), and LH/FSH ratio > 1.23 showed sensitivity of 63.2–78.2% and specificity of 84.4–93.7% in PCOS diagnosis in their sample. It is important to recognize that assays differ resulting in different androgen values. Asanidze et al. reported that around 50% of adolescents with PCOS according to Rotterdam criteria and NIH criteria have biochemical hyperandrogenism but the cut off values used were not reported in the study [68]. Adolescents with higher free androgen index at 15–16 years of age are more likely to develop PCOS at the age 26 years [48].

Despite the fact that isolated clinical hyperandrogenism and biochemical hyperandrogenism occur in 16.1% and 6.6% of adolescents, respectively, only 1.3% have both clinical and biochemical hyperandrogenism according to a large cross-sectional population study of 16 to 19 years old girls in Italy [72]. Adolescents with isolated hyperandrogenism (clinical and/or biochemical) and regular menstrual cycles should not be diagnosed with PCOS but should be considered “at risk of PCOS” (Figure 2).

### 3.3. Other Investigations and/or Features Not Part of the Diagnostic Criteria

Other investigations and/or features that are not part of the criteria for the adolescent PCOS diagnosis are included in this section. These are helpful to rule out other conditions causing menstrual cycle irregularities and/or hyperandrogenism; and/or comorbidities associated with PCOS and include blood tests, pelvic ultrasound, anti-Müllerian hormone (AMH), and insulin resistance [17,25,26]. 

Some blood tests are essential to diagnose PCOS in adolescent and adult women for the exclusion of other disorders that can cause irregular menstrual cycles and/or hyperandrogenism including beta human chorionic gonadotropin hormone (if sexually active), LH, FSH, thyroid function tests, prolactin, midnight salivary cortisol, and 17-hydroxyprogesterone (17-OHP) [79,80]. Demirci et al. investigated whether any other indicator could distinguish PCOS from non-classic congenital adrenal hyperplasia. They concluded that measuring 17-OHP was essential to differentiating between PCOS and non-classic congenital adrenal hyperplasia [81]. No differences in heterozygosity for *CYP21A2* variants was found among adolescent diagnosed with PCOS, adolescents at risk of PCOS, or healthy controls; however, one limitation of this study is that it does not appear that V281L mutation was assayed [82]. Androstenedione can be also elevated in non-classical adrenal hyperplasia. Mildly elevated DHEAS can be observed in adolescents with PCOS [58], but very high DHEAS levels are more likely to indicate the presence of an androgen secreting tumor [83,84].

#### 3.3.1. Pelvic Ultrasound to Evaluate PCOM

Even though pelvic ultrasound and PCOM are part of the Rotterdam diagnostic criteria for PCOS in adult women [20] it is not recommended for the diagnosis of PCOS during adolescence as it can cause over diagnosis of PCOS during this life stage [3,32]. This is supported by previous evidence summarized in adolescent international guidelines [17,25,30] and more recent evidence (Figure 3) [32,45,85]. Please note that the international evidence-based guidelines recommended not to use pelvic ultrasound for the diagnosis of PCOS in those with gynecological age of <8 years [17].

There are two main reasons for avoiding the use of pelvic ultrasound during adolescence. The first one is the fact that the majority of ultrasounds are made trans-abdominally not trans-vaginally, affecting the accuracy of findings [17]. There are two studies that used a trans-rectal ultrasound in adolescents. One study showed higher mean ovarian volume (9.2 vs. 4.4 cm^3^) in 69 adolescents diagnosed with PCOS according to NIH criteria compared to 26 healthy adolescents and reported that a mean ovarian volume of 6.74 cm^3^ had a 92.3% specificity and 75.4% sensitivity to distinguish PCOS in Chinese adolescents [86]. This study did not evaluate ovarian follicle count another component of PCOM. The second study showed that trans-rectal ultrasound was more reliable than trans-abdominal ultrasound evaluating PCOM but it also highlighted that healthy adolescents also had PCOM [87]. This study also showed that ovarian stromal to total area ratio was significantly higher in adolescents with PCOS compared to healthy adolescents with PCOM and without PCOM. Ovarian stromal to total area ratio was most significantly correlated with androgen levels in adolescents with PCOS [87,88]. Pelvic magnetic resonance imaging in particular for overweight adolescents with PCOS can accurate estimate ovarian stromal to total area ratio and antral follicle count; however, it is not be feasible to use this imaging modality routinely [89,90].

The second reason for avoiding using pelvic ultrasound during adolescence is the presence of PCOM in healthy adolescents, which can be a transient condition [52,85,91]. There is also significant overlap of PCOM in healthy adolescents and in adolescents with PCOS [32,45,52,71,85,91,92,93,94]. PCOM has been demonstrated in healthy adolescents using transabdominal ultrasound and irrespective of the PCOM criteria used [92]. The prevalence of PCOM according to Rotterdam Consensus (ovarian volume larger than 10 cm^3^ or more than 12 follicles [20]) was 34.3%; according to Androgen Excess-PCOS Society (ovarian volume larger than 10 cm^3^ [95]) was 25.3%; and according to the international adolescent PCOS consensus (ovarian volume larger than 12 cm^3^ [25]) was 12.8% [92]. Higher prevalence of PCOS up to 57% has been reported using Rotterdam criteria in healthy adolescents [85]. A recent cross-sectional population-based study of 257 healthy adolescents showed that PCOM with normal ovarian stromal to total area ratio is more likely to occur 1–3 years post-menarche and PCOM with increased stromal to total area ratio more likely to occur four years after menarche [85]. The presence of PCOM is higher in adolescents with irregular menstrual cycles compared to healthy girls from a population-based study. [45] Recent studies that included adolescents at least two years post menarche using Rotterdam criteria for PCOS diagnosis during adolescence demonstrated no difference in ovarian volume but higher antral follicle count between adolescents with PCOS and healthy controls [68]. In contrast, Khaschenko and Assens showed both higher ovarian volume and antral follicle count in adolescents with PCOS [45,78].

Pelvic ultrasound can be used to evaluate other possible uterine or ovarian abnormalities in adolescents that present with primary amenorrhoea [96].

#### 3.3.2. Anti-Müllerian Hormone (AMH)

AMH is a glycoprotein of the transforming growth factor beta family secreted by granulosa cells of developing ovarian follicles in females. AMH levels increase through childhood in healthy females before declining with age later in life [97,98,99,100]. AMH has been related to ovarian follicle count and it is considered a marker of ovarian reserve [45,101]. Elevated AMH levels relate to PCOM in non-obese adolescents with regular menstrual cycles [92,102].

The use of serum AMH as a single test for diagnosis of PCOS in women or adolescents is not currently recommended due to heterogeneity between studies in relation to age, assays used and PCOS diagnosis criteria used. Studies showed an important overlap in values in women with and without PCOS [17,30,103]. A review and a recent study using the international evidence-based guidelines for PCOS diagnosis in 154 adolescents support the use of AMH as an additional diagnostic marker for adolescents at risk of PCOS [68,104].

Some studies evaluating AMH in the diagnosis of adolescent PCOS have used Rotterdam criteria for PCOS diagnosis, which is not appropriate for adolescents [8,9,78,105,106,107,108]. The following studies have used PCOS NIH diagnosis criteria of irregular menstrual cycles and hyperandrogenism with inconsistent results in relation to AMH levels in adolescents with PCOS [109,110,111,112]. AMH levels are higher in non-obese [109] and obese adolescents with PCOS [111,113,114] and AMH levels decreased with weight loss and other treatments in adolescents with PCOS [68,114,115].

Few studies in adolescents have determined AMH cut off values for PCOS diagnosis with variable sensitivities and specificities, which can increase with the addition of other PCOS features such as total testosterone levels [78,111,116]. Cut offs reported included AMH values of 5.8 ng/mL (41.4 pmol/L) [9], 5.95 ng/mL (42.5 pmol/L) [109], 6.26 ng/mL (44.7 pmol/L) ([111], 6.32 ng/mL (45.1 pmol/L) [116] and 7.2 ng/mL (51.8 pmol/L) [78,110]. These cut offs are higher compared to the cut offs reported in a large cohort of women with PCOS [117]. An AMH of 3.15 ng/mL (22.5 pmol/L) at 16 years of age predicted PCOS at 26 years of age diagnosed by both NIH and Rotterdam criteria in a population-based cohort study [59]. This is in contrast to a recent longitudinal cohort study that reported that adolescent AMH levels were not a prognostic marker for PCOS in adult women [63].

AMH levels alone may not be able to be used as criteria for adolescent PCOS diagnosis but might help supporting the diagnosis if adolescents meet both irregular menstrual cycles and hyperandrogenism criteria.

#### 3.3.3. Insulin Resistance

Despite that insulin resistance as manifested by acanthosis nigricans and higher insulin levels occur commonly in adolescents with PCOS and it is exacerbated by obesity; this is not recommended for the diagnosis of PCOS during adolescence [25,26,29]. On the other hand, the presence of insulin resistance should reinforce the screening of adolescents for type 2 diabetes as a comorbidity [118]. There is a high incidence of type 2 diabetes in adolescents with PCOS [6,119] and both diabetes and PCOS increase risk of other comorbidities such as depression during adolescence [120].

Adolescents with insulin resistance and other features of metabolic syndrome require healthy lifestyle advice irrespective of PCOS diagnosis during adolescence (Figure 2).

## 4. Discussion and Conclusions

PCOS diagnosis during adolescence is more challenging and controversial due to an overlap with physiological events of puberty, which are part of the diagnostic criteria in adult women. The only criterion that applies to adolescents from all adult diagnostic criteria is the exclusion of other conditions that mimic PCOS. This review summarized the available evidence in relation to PCOS diagnostic criteria for adolescents highlighting the need for using two main criteria (NIH criteria): the first one is the presence of irregular menstrual cycles which must be well defined according to the number of years post-menarche and the second one is hyperandrogenism (clinical and/or biochemical) (Figure 3). Additionally, pelvic ultrasound and PCOM should not be used as criterion for adolescent PCOS diagnosis, which precludes the use of PCOS Rotterdam diagnostic criteria during adolescence. There is a potential for using AMH levels to support PCOS diagnosis if adolescents meet two main criteria. The research including adolescents who meet only one of the PCOS diagnostic criteria either irregular menstrual cycles or hyperandrogenism is limited at present but these adolescents should be considered “at risk of PCOS” and ongoing follow up should be established with reinforcement of healthy lifestyle (Figure 2). Longitudinal research tracking physiological events of puberty from menarche will clarify the trajectory of symptoms of adolescents “at risk of PCOS” and review if in some adolescents, we may be too early to make the diagnosis or we may be missing an opportunity for diagnosis around the time of transition of care to adult physicians.

## Figures and Tables

**Figure 1 diagnostics-12-01931-f001:**
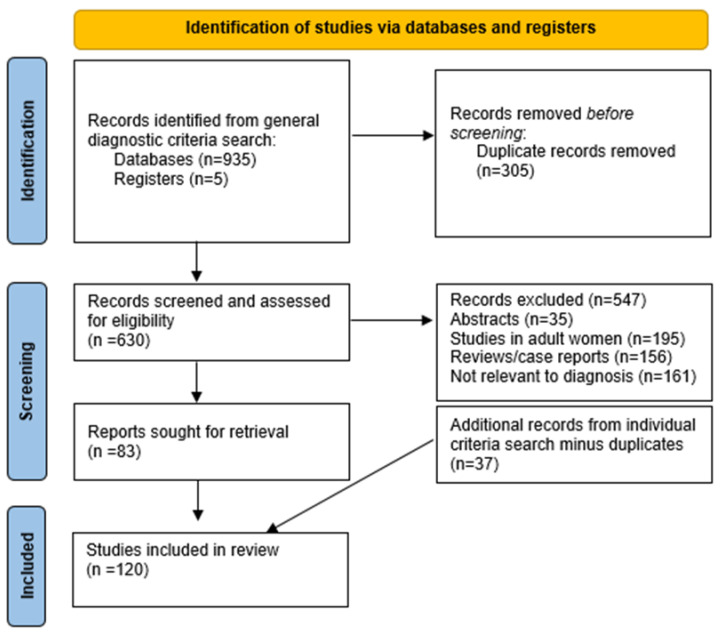
PRISMA search flow algorithm.

**Figure 2 diagnostics-12-01931-f002:**
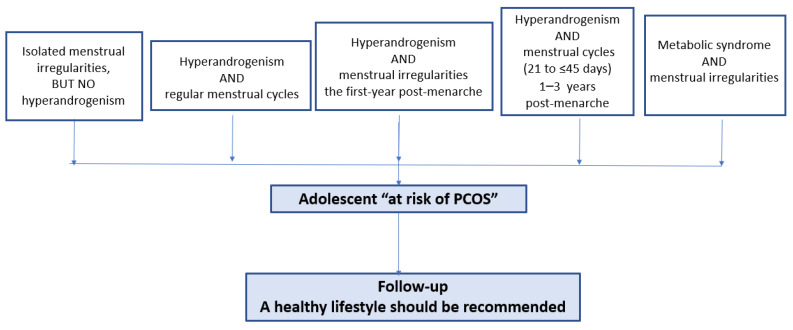
Definitions of adolescents “at risk of PCOS”.

**Figure 3 diagnostics-12-01931-f003:**
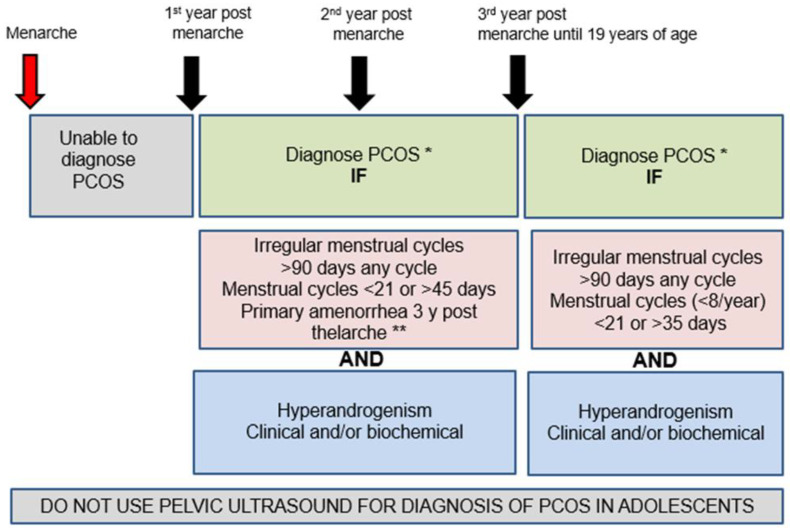
Adolescent PCOS diagnosis according to time post-menarche. * After other conditions that mimic PCOS have been excluded. ** Further investigations might be required to evaluate primary amenorrhea.

**Table 1 diagnostics-12-01931-t001:** Current specific consensus/guidelines criteria for diagnosis of PCOS during adolescence.

Criteria Definition	Witchel S et al. 2015 [25]	Ibanez L et al. 2017 [26]	Pena AS et al. 2020 [17]
Menstrual Irregularity Ovulatory dysfunction	Menstrual cycles < 20 days and >45 days two years post-menarche	Irregular cycles two years post-menarche	Strict definition according to time post-menarche Irregular cycles are normal 1st year post-menarche Menstrual cycles < 21 and >45 days 1–3 years post-menarche Menstrual cycles < 21 days and >35 days 3 years post-menarche (<8 cycles per year)
	Menstrual cycles > 90 days 1 year post-menarche	Menstrual cycles > 90 days 1 year post-menarche	Menstrual cycles > 90 days 1 year post-menarche
	Primary amenorrhea by 15 years or after 2–3 years post thelarche	Primary amenorrhea in girls that completed puberty	Primary amenorrhea by 15 years or after 3 years post thelarche
Hyperandrogenism	Clinical: moderate to severe hirsutism (no definition provided) and/or persistent acne unresponsive to topical therapy Rarely alopecia Biochemical: confirmation test in girls with hyperandrogenism Persistent elevation of total testosterone and/or free testosterone A single androgen test two standard deviations above the mean for the assay	Clinical: progressive hirsutism and/or moderate to severe acne unresponsive to topical therapy (severe cystic acne) Rarely alopecia Biochemical: confirmation test in girls with hyperandrogenism using high quality assays. No clear cut off for testosterone given	Clinical: hirsutism defined as modified Ferriman Gellway score. 4–6 and/or severe acne Rarely alopecia Biochemical: In females with irregular cycles yet without hyperandrogenism testosterone, free testosterone of free androgen index can assist with diagnosis. No cut offs given

## Data Availability

Not applicable.

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
