# Peer review of "Criteria for Diagnosis of Polycystic Ovary Syndrome during Adolescence: Literature Review"

_diagnostics, 2022, doi:10.3390/diagnostics12081931_

Round 1
Reviewer 1 Report
The authors present the literature review with the focus on the evidence related to the criteria for the diagnosis of PCOS during adolescence, and the topic is of interest.
The review is not systematic, nevertheless, the authors followed the PRISMA model in their search strategy, included all available sources of literature. Therefore, this review presents the main publications on the topic discussed.
The comments by sections are as follows:
Title and abstract. Please consider the identification of the manuscript as a literature review in the title.
The section “Search strategy”. It is reasonable to indicate all types of publications to be included in the review. Currently, there are original studies, reviews (systematic reviews with meta-analysis and just reviews, population-based studies (both in selected or unselected populations), consensus papers and others.
The section “Other investigations and/or features not part of the diagnostic criteria”. Please, add the references to support the phrase regarding DHEAS (lines 271-272).
Section “Pelvic ultrasound to evaluate PCOM”. Consider mentioning that International evidence-based guidelines (2018) recommended not to use pelvic US for the diagnosis of PCOS in those with gynaecological age of <8 years.
References. Please check if the reference #103 is correct ( In Press since the 2019??)
Author Response
Responses to Reviewer 1
The authors present the literature review with the focus on the evidence related to the criteria for the diagnosis of PCOS during adolescence, and the topic is of interest. The review is not systematic, nevertheless, the authors followed the PRISMA model in their search strategy, included all available sources of literature. Therefore, this review presents the main publications on the topic discussed.
The comments by sections are as follows:
Title and abstract. Please consider the identification of the manuscript as a literature review in the title.
Response: Title has been changed to Criteria for diagnosis of polycystic ovary syndrome during adolescence: Literature review
See Page 1, line 3.
The section “Search strategy”. It is reasonable to indicate all types of publications to be included in the review. Currently, there are original studies, reviews (systematic reviews with meta-analysis and just reviews, population-based studies (both in selected or unselected populations), consensus papers and others.
Response: The following sentence has now been added to the search strategy. See Page 2, Lines 88-90
This review included original studies in adolescents, systematic reviews and meta-analysis, population-based studies (both in selected and unselected populations), consensus papers and international guidelines.
The section “Other investigations and/or features not part of the diagnostic criteria”. Please, add the references to support the phrase regarding DHEAS (lines 271-272).
Response: The following references have been added to support this phrase. See Page 6, line 230 and Page 36, lines 1496-1499.
- Khan SH et al. Dehydroepiandrosterone Sulfate (DHEAS) Levels in Polycystic Ovarian Syndrome (PCOS) J Coll Physicians Surg Pak. 2021 Mar;31(3):253-257. doi: 10.29271/jcpsp.2021.03.253.
- Turco AF et al. 11-Oxygenated androgens in health and disease. Nat Rev Endocrinol. 2020 May;16(5):284-296. doi: 10.1038/s41574-020-0336-x. Epub 2020 Mar 16.
Section “Pelvic ultrasound to evaluate PCOM”. Consider mentioning that International evidence-based guidelines (2018) recommended not to use pelvic US for the diagnosis of PCOS in those with gynaecological age of <8 years.
Response: The following sentence “Of note the international evidence-based guidelines recommended not to use pelvic ultrasound for the diagnosis of PCOS in those with gynecological age of <8 years” has been added to the section Pelvic ultrasound to evaluate PCOM. See Page 6, lines 287-289.
References. Please check if the reference #103 is correct (In Press since the 2019??)
Response: Thanks for comment - Reference has now been updated. See Page 37, line 1483.
Teede H, Misso M, Tassone EC, Dewailly D, Ng EH, Azziz R, Norman RJ, Andersen M, Franks S, Hoeger K, Hutchison S, Oberfield S, Shah D, Hohmann F, Ottey S, Dabadghao P, Laven JSE. Anti-Müllerian Hormone in PCOS: A Review Informing International Guidelines. Trends Endocrinol Metab. 2019 Jul;30(7):467-478. doi: 10.1016/j.tem.2019.04.006. Epub 2019 May 31.
Reviewer 2 Report
In this interesting review, the author tried to review the diagnosis of adolescence PCOS. Below some comments regarding the manuscript. 1. The manuscript is well writing. 2. Several points regarding hyperandrogegism and hirsutism should be insisted: a. Hirsutism must be distinguished from hypertrichosis, which is defined as excessive vellus hair distributed in a non-sexual pattern. b. Mild hirsutism may not be a sign of hyperandrogenemia. c. It is very difficult to define mild hirsutism in adolescence PCOS, as Ferriman-Gallwey scoring system may not be suitable for adolescence PCOS. 3. Regarding the menstrual irregularity, Pena et al (BMC Medicine 2020;18:72) suggested "irregular menstrual cycles defined according to years post-menarche; > 90 days for any one cycle (> 1 year post-menarche), cycles< 21 or > 45 days (> 1 to < 3 years post-menarche); cycles < 21 or > 35 days (> 3 years post-menarche) and primary amenorrhea by age 15 or > 3 years post-thelarche."Author Response
Responses to Reviewer 2
In this interesting review, the author tried to review the diagnosis of adolescence PCOS. Below some comments regarding the manuscript.
1. The manuscript is well writing. Thanks for comment.
2. Several points regarding hyperandrogenism and hirsutism should be insisted: a) Hirsutism must be distinguished from hypertrichosis, which is defined as excessive vellus hair distributed in a non-sexual pattern.
Response: Thanks for comment. The following sentence has been added: Hirsutism must be distinguished from hypertrichosis, which is defined as excessive vellus hair distributed in a non-sexual pattern. See Page 5, lines 212-213.
b) Mild hirsutism may not be a sign of hyperandrogenemia and c) It is very difficult to define mild hirsutism in adolescence PCOS, as Ferriman-Gallwey scoring system may not be suitable for adolescence PCOS.
Response: We agree with the reviewer that no studies accurately and consistently define the optimal cut point for hirsutism in adolescents as per submitted manuscript (Page 4, lines 195-196). We have also now added the following sentence “and mild hirsutism may reflect ethnic variation or normal pubertal progression rather than indicating hyperandrogenemia during adolescence.” Nevertheless, the cutaneous findings need to be interpreted within the clinical context of a specific patient. See Page 4, lines 196-199.
3. Regarding the menstrual irregularity, Pena et al (BMC Medicine 2020;18:72) suggested "irregular menstrual cycles defined according to years post-menarche; > 90 days for any one cycle (> 1 year post-menarche), cycles< 21 or > 45 days (> 1 to < 3 years post-menarche); cycles < 21 or > 35 days (> 3 years post-menarche) and primary amenorrhea by age 15 or > 3 years post-thelarche."
Response: Thanks for comment. We have now added this sentence “In addition, if a menstrual cycle is longer than 90 days after one year post-menarche is also a sign of menstrual irregularity (Table 1, Figure 3) [Pena et al BMC Medicine 2020;18:72].
See Pages 3, Lines 138-139.